# The Evolutionary History of Vertebrate Adhesion GPCRs and Its Implication on Their Classification

**DOI:** 10.3390/ijms222111803

**Published:** 2021-10-30

**Authors:** Aline Wittlake, Simone Prömel, Torsten Schöneberg

**Affiliations:** 1Division of Molecular Biochemistry, Rudolf Schönheimer Institute of Biochemistry, Medical Faculty, Leipzig University, 04103 Leipzig, Germany; a.j.wittlake@gmx.de; 2Department of Biology, Institute of Cell Biology, Heinrich Heine University Düsseldorf, 40225 Düsseldorf, Germany

**Keywords:** adhesion GPCR, G protein coupled receptor, nomenclature, phylogeny, evolution

## Abstract

Adhesion G protein-coupled receptors (aGPCRs) form a structurally separate class of GPCRs with an unresolved evolutionary history and classification. Based on phylogenetic relations of human aGPCRs, nine families (A–G, L, V) were distinguished. Taking advantage of available genome data, we determined the aGPCR repertoires in all vertebrate classes. Although most aGPCR families show a high numerical stability in vertebrate genomes, the full repertoire of family E, F, and G members appeared only after the fish–tetrapod split. We did not find any evidence for new aGPCR families in vertebrates which are not present in the human genome. Based on ortholog sequence alignments, selection analysis clearly indicated two types of tetrapod aGPCRs: (i) aGPCR under strong purifying selection in tetrapod evolution (families A, B, D, L, V); and (ii) aGPCR with signatures of positive selection in some tetrapod linages (families C, E, G, F). The alignments of aGPCRs also allowed for a revised definition of reference positions within the seven-transmembrane-helix domain (relative position numbering scheme). Based on our phylogenetic cluster analysis, we suggest a revised nomenclature of aGPCRs including their transcript variants. Herein, the former families E and L are combined to one family (L) and GPR128/ADGRG7 forms a separate family (E). Furthermore, our analyses provide valuable information about the (patho)physiological relevance of individual aGPCR members.

## 1. Introduction

With more than 800 genes G protein-coupled receptors (GPCRs) form the most abundant superfamily in the human genome [1,2]. Based on phylogenetic sequence relations, GPCRs have been grouped into five classes: Glutamate, Rhodopsin, Adhesion, Frizzled, and Secretin receptors, the so-called GRAFS classification [3,4]. The NC-IUPHAR classification considers also non-vertebrate receptors and sorts GPCRs into class A (rhodopsin-like), class B (adhesion- and secretin-like), class C (metabotropic glutamate receptor–like), class D (fungal mating/pheromone receptor–like), class E (cyclic AMP receptor–like), and class F (frizzled/smoothened–like) [5]. GPCRs participates in almost every physiological function by mediating the signal transduction of photons, ions, neurotransmitters, metabolites, hormones, and odors. The class of adhesion GPCRs (aGPCRs) are also involved in transducing mechanical forces [6,7,8] and in cell–cell and cell–matrix interactions [9]. This still under-investigated class contains 33 mammalian receptor homologs, most of them with unknown physiological properties [9]. Adhesion GPCRs are equipped with adhesive structural folds (e.g., leucin-rich domain, Ig domain, pentraxin domain) and a G protein-coupled receptor Autoproteolysis-INducing (GAIN) domain in their very large extracellular N termini [10]. These are anchored to the plasma membrane via a seven-transmembrane helices (7TM) domain, which shows some structural resemblance to the 7TM domain of the secretin-like receptor class [11]. However, the phylogenetic relation between aGPCRs and secretin-like receptors is still unsolved. There is some support for a common ancestry [12,13] but also evidence for a descent of secretin-like receptors from aGPCRs [11,14,15]. Recently, a consortium of scientists working on aGPCRs suggested a unified nomenclature [9]. Based on the phylogeny of the human aGPCR genes, nine families (previously defined as “subfamilies”) (ADGRA, B, C, D, E, F, G, L, V) were defined (Figure 1).

Thereby, the phylogenetic relations were determined mainly on the basis of the 7TM domain amino acid sequences of human aGPCRs [19], because their extracellular N termini are highly variable in length and an evolutionary result of combinatory domain rearrangements. In a recent study, we suggested to revise this nomenclature (version 2.0) as the hierarchical organization of aGPCRs, and GPCRs in general, contains several ambiguities and inconsistencies [15]. The growing number of solved genomes from diverse species revealed that the defined receptor family structure is not fully supported by phylogenetic analyses. Therefore, we suggested a modified classification based on phylogenetically supported levels (level 1 to level 6). Level 1 (species) defines the individual subtypes (e.g., zebrafish LPHN1a/ADGRL1a and LPHN1b/ADGRL1b), level 2 (genus) combines closely related subtypes of latrophilin-1 (e.g., all LPHN1/ADGRL1), and level 3 (family) combines then all latrophilins (e.g., all LPHN/ADGRL1-4). The entity of level 4 (order) is currently undefined. In case of class B receptors, it may distinguish between aGPCRs and secretin-like GPCRs. Level 5 (class) combines all aGPCRs and secretin-like GPCRs to one class and level 6 represents the phylum of GPCRs.

Still, the central question of such classification remains: What determines a ‘level’ in this hierarchy? One parameter could be the significant clustering of receptors within phylogenetic trees. Thus, we performed an extensive analysis of the evolutionary history of vertebrate aGPCRs and secretin-like GPCRs and tested whether such level-based nomenclature allows for unbiased classification. The consequent application of cluster-based level definition led to a further revised and refined classification of vertebrate aGPCRs (version 3.0). The new nomenclature also considers the multitude of transcript variants aGPCR genes can encode. Numerical variabilities of defined domains within the N terminus, such as EGF domains, are frequent and have been described, for example, in EMR2/ADGRE2 and CD97/ADGRE5 [20,21]. This is of high relevance because the derived proteins can substantially differ in structure and function [22]. Indeed, splice variants showing significant functional differences were identified for GPR56/ADGRG1 [23], latrophilins [24,25] and GPR116/ADGRF5 [22]. There is even strong evidence for alternative promoters within aGPCR genes generating transcript variants encoding N-terminally truncated aGPCRs [22]. Therefore, a clear denomination of such transcript and protein variants is of high importance for the aGPCR field. Furthermore, our assembled vertebrate repertoire of aGPCRs allowed for structural comparison, revealed structural determinants relevant for maintaining the specific functions of vertebrate aGPCRs, and helped to define reference positions within the 7TM domain (relative position numbering scheme). Based on the identification of signatures of positive selection and frequency analysis of loss-of function variants in humans, we evaluated the possible functions of aGPCRs in adaptive processes and their role in human diseases.

## 2. Results and Discussion

### 2.1. Repertoire of aGPCRs in Mammalian Orders

The current classification of aGPCRs comprises nine families, all found in the human genome (Figure 1). However, not all family members are functional in humans. For example, the open reading frame of EMR4/ADGRE4 is interrupted by a frameshifting deletion, implying that ADGRE4 is a pseudogene in humans [26]. The question remains whether all mammals are equipped with the same repertoire of aGPCRs. A gain or loss of aGPCRs can be invaluable information to interpret the functional relevance of specific members of the receptor class. Some numerical differences of aGPCRs are already known. For example, ADGRD2/GPR144 is not present in mouse [15] and GPR111/ADGRF2 and GPR115/ADGRF4 are absent in bottlenose dolphin (*Tursiops truncatus*) genome [27].

To address this question, we now systematically analyzed the presence of all nine aGPCR families in the available mammalian genomes by searching public databases (see Methods). In most cases, more than 100 mammalian orthologs per aGPCR family could be retrieved from NCBI and Ensemble using string- and sequence-based search strategies. Following alignments with MUSCLE and manual curation, the 7TM domain-encoding sequences were utilized to build NJ trees and to assign the entries to one of the existing aGPCR family (Suppl species.fasta). Sequences with a one-to-one orthology to the human aGPCR repertoire were found in at least one other species of the main mammalian lineages (Monotremata, Marsupialia, and Eutheria: Atlantogenata, Boreoeutheria) (Table 1, Appendix A). This indicates that all nine aGPCR families were already introduced in the vertebrate genome before mammals arose more than 178 million years ago (mya) [28]. We did not find aGPCRs in mammals that cluster independently of the known families and form an additional family.

However, several aGPCR members show gene duplications during mammalian evolution. For example, at least two EMR2/ADGRE2 paralogs exist in many mammals like in Felidae, Carnivora, Marmotini, and Artiodactyla (Appendix A). The little brown bat (*Myotis lucifugus*) even contains four paralogous sequences of EMR2/ADGRE2 in its genome. As another example, EMR4/ADGRE4 underwent gene duplication in early eutherian evolution but only one copy was kept in primates and rodents. Multiple copies of ADGRE4 are found in the genomes of the African elephant (*Loxodontus africanus*) and the platypus (*Ornithorhynchus anatinus*).

Interestingly, two members of the aGPCR class show a high frequency of pseudogenes across the mammalian linages: EMR4/ADGRE4 and GPR144/ADGRD2 (Table 1), indicating specific functions in some mammalian species, but not a vital requirement of these two receptors in mammals. The previously identified loss of GPR111/ADGRF2 and GPR115/ADGRF4 in the dolphin genome [27] seems to be representative for an absence of GPR111/ADGRF2 in all sequenced extant *Cetaceans* and of GPR115/ADGRF4 in some extant toothed whales (*Odontoceti*). The high genomic dynamics of these aGPCR members reflected by gene gain and loss may contribute to specific adaptation or environment-related loss-of-constraints.

### 2.2. Repertoire of aGPCRs in Vertebrate Classes

We next extended our analysis of the origin of aGPCR families to all bony vertebrate classes. For unbiased retrieval of all aGPCR-related sequences from representative species (bony fish, amphibians, reptiles, birds, mammals), we mined sequence databases with a sophisticated sequence search strategy (see Materials and Methods). Extracted sequences were aligned and assigned to the aGPCR families. As shown in Figure 2 and Appendix A, all aGPCR families have at least one assigned fish ortholog, indicating that at least one member of all aGPCR families already existed in Silurian vertebrates about 419 million mya [30]. However, a fish-mammal one-to-one orthology for every member of an aGPCR family is found only for families A, B, C, D, L, and V. In some of these families, individual species lack a member (e.g., some birds lack LPHN1/ADGRL1, GPR133/ADGRD1 is missing in the green anole (*Anolis carolinensis*), GPR144/ADGRD2 is missing in some mammals) (Appendix A). However, we cannot exclude that the current genome assemblies just lack these sequences. In contrast, there is no clear one-to-one orthology assignment possible in the E, G, and F families (Figure 2, Appendix A). An interesting finding is that, although the mammalian CD97/ADGRE5 has orthologs in fishes, amphibians, and reptiles, CD97/ADRGE5 is fully missing in all birds investigates and in platypus (Figure 3). The lack of CD97/ADRGE5 in all available bird genomes almost exclude coverage or assembly problems. Furthermore, GPR128/ADGRG7 lacks orthologs in all reptiles investigated and in some amphibians and birds (Appendix A). For the group containing EMR1-4/ADGRE1-4, related sequences can be found in amphibians and reptiles but rarely in birds (Appendix A). However, a clear phylogenetic assignment of the individual members of the ADGRE1-4 group is only possible in mammals. This suggests that both receptors, ADRGE5 and the common ancestor of the ADGRE1-4 group, have diverged mainly after tetrapod split from fishes in Devonian [31].

The ADGRF family is a cluster of aGPCRs, in which GPR113/ADFRF3 evolved separately from the others with a clearly assigned one-to-one orthology from fish to mammals. The members GPR110,111,115,116/ADGRF1,2,4,5 have common orthologs in fish, but most probably diverged after the fish–tetrapod split. GPR116/ADRGF5 has a well-resolved one-to-one orthology in tetrapods. However, GPR110,111,115/ADGRF1,2,4 have several common orthologs in amphibians, reptiles, and birds but a clear one-to-one orthology was found only after the split of mammals from the other tetrapods (Appendix A).

Gene duplication of aGPCR members leading to paralogs was frequently observed in fish, most probably because of a teleost-specific genome duplication about 320 mya [33]. For example, BAI1/ADGRB1, CELSR1/ADGRC1, CD97/ADRGE5, GPR64/ADGRG2, GPR112/ADGRG4, LPHN2/ADGRL2, and LPHN3/ADGRL3 have at least two orthologous sequences in zebrafish (Appendix A). In some cases, there is radiation of aGPCR members in fishes, amphibians, or reptiles. For example, ADGRF-like receptors radiate in zebrafish and several amphibian species (Appendix A). Similarly, ADGRE-like receptors radiated in zebrafish, as well as several amphibian and turtle species.

In sum, our analyses revealed a high stability of most aGPCR families in mammalian genomes. Most aGPCR families also show a high numerical stability in vertebrate genomes, but the full repertoire of family E, F, and G members appeared only after the fish–tetrapod split.

### 2.3. Clustering of aGPCRs Based on Vertebrate Orthologs

The current nomenclature of aGPCRs is mainly based on the phylogenetic relation of human aGPCRs [19]. Although there were no stringent cut-off criteria that defined an aGPCR family, significant phylogenetic clustering of individual human receptors has led to the current nomenclature. Indeed, families A, B, C, D, F, and V form individual sequence clusters, which are supported by bootstrap tests and with different alignment algorithms (Figure 2, Appendix A).

However, a number of inconsistencies already encouraged us in a previous study [15] to suggest that the ADGRG family should be split into three families/groups: GPR56,97,114/ADGRG1,3,5; GPR64,112,126/ADGRG2,4,6; and GPR128/ADGRG7. As supported by our new analyses (Figure 2, Appendix A), vertebrate GPR128/ADGRG7 evolved separately and do not cluster with any other ADGRG member, thus forming a separate family. GPR56,97,114/ADGRG1,3,5 and GPR64,112,126/ADGRG2,4,6 form one cluster, which is supported by bootstrap tests. However, the two groups present with different evolutionary dynamics. Although the GPR64,112,126/ADGRG2,4,6 group shows a one-to-one orthology in all investigated vertebrates, members of the GPR56,97,114/ADGRG1,5,3 group have common fish orthologs, but GPR56/ADGRG1 and GPR114/ADGRG5 most probably separated in tetrapods. Interestingly, the presence of GPR56,97,114/ADGRG1,3,5 is instable in amphibians and reptiles. These findings resemble the evolutionary history of the families ADGRE and ADGRL. Both families significantly cluster (Figure 2) and ADGRL evolved evolutionarily conservative whereas ADGRE is numerically instable (see above).

It should be noted that we did not find any new aGPCRs in vertebrates not related to the already known aGPCR families in the human genome. The ability of our pipeline to identify new members or families is demonstrated below (2.4), where we found new families in chordates. However, the inconsistencies between the defined ADGRG, ADGRL, and ADGRE families and their phylogenetic clustering suggest a revision of the current nomenclature of vertebrate aGPCRs.

### 2.4. The Evolutionary Dynamics of Vertebrate aGPCRs and Its Implication on Their Nomenclature

The historical denomination of GPCRs was mainly based on their agonists, ligands, or physiological functions. This led to unstructured naming of receptors. Constant work of the Nomenclature Committee of the Union of Basic and Clinical Pharmacology (NC-IUPHAR) attempts to synchronize GPCR nomenclature based on sequence homology and pharmacological properties [2]. This is especially important for so-called orphan GPCRs, which were identified from sequenced genomes and transcriptomes and where no agonist and physiological function are known yet. However, the historically evolved and current nomenclature of GPCRs is neither entirely systematic nor logical, even for those GPCRs with known endogenous agonists and signal transduction. Since ligand- or signal-transduction-based classifications of GPCRs are not straightforward, we have recently argued that hierarchy- and phylogeny-based ordering systems, such as the GRAFS system [4] and the level system [34], provide the best resolution; however, they fall short in non-rhodopsin classes due to a number of inconsistencies and lack of ordering parameters [15]. Such nomenclature issues are not new and were addressed for several protein families with different evolutionary histories [35,36].

Therefore, we have suggested a level-based ordering hierarchy [15] by keeping the previously established ADGR denomination [9,19]. The level system follows a bottom-up ordering logic in the phylogenetic classification of GPCRs. This system uses hierarchy levels denominated by taxonomical terms, which distinctly separate species (level 1), genus (level 2), family (level 3), order (level 4), class (level 5), and phylum (level 6) (Figure 4). Taking advantage of our in-depth phylogenetic analyses of aGPCRs and secretin-like receptors, we can now assign aGPCRs based on amino acid sequence alignments of the 7TM domain and bootstrap-supported phylogenetic analyses (Figure 2) to the level system (Figure 4) and provide a revised nomenclature of aGPCRs (Table 2). The following parameters were defined to assign aGPCRs to the different levels:(1)Phylogenetic analyses based on an amino acid sequence alignment using representative aGPCRs of all vertebrate classes.(2)Significant clustering in bootstrap analyses (≤50%) defines the hierarchic level.(3)Adhesion GPCRs and secretin-like GPCRs form a separate class (level 5, class) compared to other the GPCR classes.(4)Although the secretin-like class clusters within the aGPCR class and, therefore, should follow the same nomenclature rules as the aGPCR, we pragmatically decided to keep the secretin-like GPCRs and the aGPCRs as two separate orders (level 4). The aGPCR order is abbreviated with ‘ADGR’.(5)Level 3 (family) is defined only when clustering supports family formation. The family is abbreviated with a single upper letter, e.g., ‘ADGRF’.(6)Level 2 (genus) is defined only when clustering supports direct orthology in fishes and in mammals. Level 2 is abbreviated with a number, e.g., ‘ADGRF2′. The continuous numbering systematically follows their phylogenetic relation.(7)Level 1 (species) is the individual receptor in a given species. Level 1 is abbreviated with a lower character, e.g., ‘ADGRF2a’ preferable following their phylogenetic relation.(8)mRNA splice variants of the same gene should be labeled with a period and a continuous number, e.g., ‘ADGRF2a.1′.

Consequent application of this nomenclature leads to the following changes compared to the previous standards: (i) secretin-like GPCRs and aGPCRs are members of one class but form separate orders (level 4) (Figure 4), (ii) the former subfamilies or groups are now termed ‘families’ (Figure 4), (iii) the former subfamilies ADGRE and ADGRL now form the common family ‘ADGRL’ (Table 2), (iv) GPR98/ADGRG7 now forms the separate family ‘ADGRE’ (Table 2), (v) splice variants are now included in the nomenclature, and (vi) several members do not show a one-to-one orthology between fishes and mammals and, therefore, where considered as level 1 (e.g., the ADGRF family supports only two genus (level 2) renaming ADGRF1,2,4,5 to ‘ADGRF2a,b,c,d’—Figure 4). Because not all aGPCRs have a one-to-one orthology within all vertebrate classes, it is impossible to derive a one-to-one orthology at the level 1 assignment. For example, the human ADGRF2a must not be the ortholog of the zebrafish ADGRF2a. The lower character at the end of the aGPCR name is only individually to the animal species. We have observed many cases where an aGPCR underwent duplication in a single species or a distinct clade but not in other vertebrates.

To distinguish such duplicated genes within a species, this lower character became necessary and, therefore, is private for the species or clade. Therefore, the abbreviation of an aGPCR gene which includes level 1 should be always given with a species abbreviation. However, there is no abbreviation systematics for species names—we simply used two letters derived from their taxonomic names (e.g., hs = *Homo sapiens*). Surely, this needs revision since several other species even have the same first two characters in the generic name and specific name, e.g., *Homo sapiens* and *Homalopoma sanguineum* (a sea snail species). Here, one has to wait for an international regulation which can then be applied to the individual aGPCRs.

### 2.5. The Origin of aGPCR Families

Our analyses above revealed that 19 aGPCRs have a fish-mammal one-to-one orthology (groups A1-3, B1-3, C1-3, D1,2, L1-4, V1, G2,4,6) and 2 of each of the families E, F, and G1,3,5. This indicates that the repertoire of the 33 human aGPCRs evolved from at least 25 ancient aGPCRs that already existed in the genome of first vertebrates. Previous analyses showed that aGPCRs are among the oldest GPCR classes being present in single-celled eukaryotes [11]. The repertoire of aGPCRs in invertebrates is significantly smaller than in vertebrates with five members in *Drosophila melanogaster* [15] and three members in *Caenorhabditis elegans* [37]. Therefore, we asked whether all vertebrate aGPCR families already occur in primitive Chordata such as Hyperoartia (lamprey, *Petromyzon marinus*), Cephalochordatae (lancelet, *Branchiostoma belcheri*), and Tunicata (*Ciona intestinalis*). For the aGPCR families A, B, C, G, and L, there was sequence evidence that these five families have at least one member in all primitive Chordata investigated (Figure 5). The ADGRD family is present in lamprey and *C. intestinalis* but not in lancelet, indicating that this family probably got lost in this species. VLGR1/ADGRV1 is not present in *C. intestinalis* and ADGRF is found only in lamprey. The ADGRE family is not found in the three primitive Chordata and, therefore, is the evolutionarily ‘youngest’ aGPCR family. Interestingly, we found two currently not assigned aGPCR families in *C. intestinalis* and lancelet (referred to as ADGRN and ADGRX, Figure 5), which are not present in vertebrates. Both families show significant radiation (up to 20 individual receptors) and cluster separately from all vertebrate aGPCR families (Figure 5). In previous studies [11,15], we and others found aGPCR sequences in Protostomia (insects, mollusks, worms) and Deuterostomia (Chordata, Hemichordate, Echinodermta), indicating an evolutionary age of a broad aGPCR repertoire as old as Bilateria. Prototypes of aGPCRs but not of secretin-like GPCRs are found in unicellular eukaryotes indicating that aGPCRs are most probably the most ancient receptors among class B GPCRs [11].

### 2.6. Secretin-Like Receptors Descended from aGPCRs by Partial Transmembrane Domain Rearrangement

Our current results strongly support previous studies with different data sets [11,14,15] that the class of secretin-like receptors descended from the aGPCR class. Therefore, we included all secretin-like GPCRs of the investigated Chordata species into our phylogenetic analysis. We clearly found close phylogenetic relations to GPR144/ADGRD2 (Figure 5). Since most secretin-like receptors and GPR144/ADGRD2 have orthologs in primitive Chordata (lamprey, lancelet, *Ciona intestinalis*), the split between GPR144/ADGRD2 and secretin-like GPCRs must have occurred before the origin of the chordates. Indeed, previous analyses showed the parallel existence of adhesion- and secretin-like GPCRs in Chordata and Echinodermata [15,41]. However, the positioning within phylogenetic trees did not always link secretin-like receptors to the ADGRD family [15]. We therefore speculated that secretin-like receptors may have emerged from rearrangements or recombination of different aGPCR families. Thus, we performed phylogenetic analyses of the 7TM domain in comparison to parts of the 7TM domain (Appendix A). The TM6-7 part of secretin-like receptors displayed some phylogenetic relations to the corresponding TM part of the ADGRD family, whereas the TM1-2 and TM3-5 fragments had higher homology to the corresponding part of other aGPCRs (Appendix A). This may indicate that secretin-like GPCRs have evolved from parts of the 7TM domain of different aGPCRs, most probably by genomic recombination.

### 2.7. Identification of Highly Conserved Residues within the 7TM Domains of aGPCRs and Secretin-Like GPCRs

Since the secretin-like receptors may have descended from aGPCRs in early animal evolution, the recently solved cryo-electron microscopy (cryo-EM) and crystal structures of the 7TM domains of the aGPCR GPR97/ADGRG3 [42] and secretin-like GPCRs [43,44,45], respectively, offer useful structural templates for homology modeling and three-dimensional studying of the 7TM domain regions of other aGPCRs. Thus, homologous residues with possible importance for ligand binding and G-protein coupling can be mutationally addressed and compared, an approach frequently used also in other structure–function relationship studies with GPCRs. However, the cryo-EM structure of GPR97/ADGRG3 exposed several significant differences between secretin-like GPCRs and aGPCRs in respect to the length, kinks, and relative orientation of TM helices [42]. For example, the cryo-EM structure of GPR97/ADGRG3 highlights W^6.55^ (referred to the new reference position L^6.50^, Figure 6A) as ‘toggle switch’ residue important for receptor activation which is missing in secretin-like receptor. Furthermore, the positioning of a proline in TM6, which causes kinking of helixes, is well-preserved secretin-like receptors but not in aGPCRs (see Figure 6A, and alignments in the provided fasta files). In contrast to GPR97/ADGRG3, members of the ADGRB, D, and F families have this proline, indicating significant differences in the helix architecture between aGPCR and supporting the phylogenetic relation between some aGPCRs and secretin-like receptors also on the structural level.

To allow comparison between the residues at different positions in the 7TM domain of different GPCRs within the rhodopsin-like class, residues are numbered according to the Ballesteros–Weinstein numbering scheme [46], where the single most conserved residue in each TM helix is designated X.50 [47]. Detailed inspection of aGPCR sequences did not reveal any motifs being 100% conserved in the 7TM domain. Already, the identification of reference positions within the individual TM domains turned out to be difficult. Considering only human sequences, the reference positions in aGPCRs and secretin-like receptors have been defined [48,49] with TM1: S^1.50^ (84.7% conservation, considering only human sequences), TM2: H^2.50^ (58.7% conservation), TM3: E^3.50^ (77.6% conservation), TM4: W^4.50^ (64.7% conservation), TM5: N^5.50^ (87.1% conservation), TM6: G^6.50^ (73.1% conservation), and TM7: G^7.50^ (Figure 6A, shown in blue). However, only a few of these assigned residues are fully conserved within aGPCRs (Figure 6A) and, overlaying the three-dimensional structures of the other GPCR classes, the respective X.50 positions are not at all structurally homologous. Therefore, we reassigned relative positions within the 7TM of aGPCRs and secretin-like GPCRs based on alignment of all human receptors.

There are only three positions—W^3.50^, P^4.50^, and G^7.50^—which are highly conserved (>95%) among human aGPCRs and secretin-like receptors and, therefore, fulfill the requirements to serve as reference positions. Considering both, human aGPCRs and secretin-like GPCRs, one should define the reference positions by best conservation: **TM1**: L^1.50^ (90.1% conserved), **TM2**: N^2.50^ (78.6% conserved), **TM3**: W^3.50^ (99.5% conserved), **TM4**: P^4.50^ (99.5% conserved), **TM5**: N^5.50^ (87.1% conservation), **TM6**: L^6.50^ (78.6% conserved), and **TM7**: G^7.50^ (96.0% conserved) (Figure 6A,B).

Next, we analyzed only vertebrate aGPCRs and the alignment exposed additional 7TM domain positions with high conservation among aGPCRs: TM3: H^3.41^, and L^3.45^, and TM6: W^6.53^ (Figure 6B, shown in magenta). Besides the disulfide bridge-forming cysteines, there are only a few other residues in the extracellular loops that are highly conserved: ECL2: Y^4.69^ (in GPR97/ADGRG3) and W^4.84^ (in GPR97/ADGRG3). When aGPCRs of all families with one representative of each vertebrate class (Figure 2) were considered and setting the cut-off to 80% conservation, three signature sequence motifs of aGPCRs can be assigned: **TM3**: LHxxxLxxFxW^3.50^, **TM4**: GxGxP^4.50^, and **TM7**: FxxxxxxQG^7.50^ (x stands for any amino acid).

We, and later others, have shown that an internal sequence, called *Stachel* sequence, serves as a tethered agonist upon activation of the receptor [50,51]. It has been proposed that during activation, this sequence is exposed or isomerizes into an active conformation that interacts with its 7TM domain binding site [52]. Peptides derived from this sequence can activate aGPCRs and show cross-reactivity between different aGPCR members and families [53]. This indicates that the binding site is at least in parts conserved between different aGPCRs and one can speculate that some of the conserved positions participate in *Stachel* binding and transduction of its intramolecular signals. Specifically, those residues, which are located in the extracellularly oriented part of the 7TM domain and which are very conserved, such as W^6.55^ and F^7.42^, may participate in *Stachel* binding. Since the core sequence of the *Stachel* is very hydrophobic [53,54], it is very likely that the binding pocket is further composed of conserved hydrophobic amino acid clusters in the extracellular oriented parts of TM5, TM6, and TM7 and the extracellular loops. Future crystallographic and cryo-EM studies, which include parts of the N terminus in their structure, will shed light on these interactions.

In sum, considering aGPCRs and secretin-like receptors as phylogenetically related GPCRs, which share conserved determinants within the 7TM domain, the interpretation of their three-dimensional structures from crystallography and cryo-EM studies and of mutagenesis data requires proper alignment of their amino acid sequences. The proposed relative numbering system (Figure 6A) may serve as a scaffold for such comparative analyses.

### 2.8. Selection on aGPCRs and Its Implication on Their Physiological Relevance

Gene duplication and even radiation was observed for many members of vertebrate aGPCRs in our study. After gene duplication, the resulting homolog can have two fates, (i) pseudogenization or (ii) gain of new function. In the latter case, one copy may accumulate mutations and acquire unique functionality without risking the fitness of the organism, which is ensured by the other homolog. To screen for signatures of selection of individual aGPCR members, the webtool Selectome was used [29]. This analysis is based on individual aGPCR ortholog alignments and trees and uses the branch-site model to determine ω-values among branches. The dN/dS ratio (or ω-value) is the ratio of the rate of non-synonymous substitutions (dN) to the rate of synonymous substitutions (dS) in codons, which can be used as an indicator of selective pressure acting on a protein-coding gene. If dN/dS < 1 one can assume negative/purifying selection, if dN/dS ≥ 1 one can assume positive selection.

As expected, most significant ω-values were found in fish aGPCR orthologs (Table 1), where after duplication one branch evolved under positive selection, whereas the other homolog remained under purifying (negative) selection. There were also a few cases in tetrapods, in which gene duplication was followed by positive selection of one copy. For example, EMR2/ADGRE2 paralogs showed signature of positive selection in the Marmotini linage. However, we also found signatures of positive selection in branches not related to gene duplication. For example, CELSR1/ADGRC1, CELSR2/ADGRC2, GPR116/ADGRF5, and GPR126/ADGRG6 show selection in mammalian branches after splitting from other tetrapods. More specific branch selection was found for EMR1/ADGRE1 in some primates (Cercopithecidae) and bears (Ursus) and EMR2/ADGRE2 in Lemuriformes and Panthera. In bird and reptile branches, CELSR2/ADGRC2, GPR116/ADGRF5, GPE56/ADGRG1, and GPR114/ADGRG5 showed significant signatures of positive selection (Table 1).

Although most aGPCR families contain some examples of positive selection in bony fishes, our analysis clearly indicates two types of tetrapod aGPCR families: (i) aGPCRs under strong purifying (negative) selection in tetrapod evolution (ADGRA, ADGRB, ADGRD, ADGRL, ADGRV); and (ii) aGPCRs with signatures of positive selection in some tetrapod linages (ADGRC, ADGRE, ADGRG, ADGRF) (Table 1). They may indicate that members of type (i) mainly participate in maintaining conserved physiological function and of type (ii) contribute to adaptive functions (e.g., environment, immune response).

Our analyses revealed that the overall repertoire of aGPCRs is very constant, not only in mammals but also in vertebrates. The families ADGRA, B, C, D, L, and V are conservative and integral parts of vertebrate genomes (Table 1). Members of these families are represented with a solid one-to-one orthology in all vertebrate classes and also found in early chordates, but with less members (Table 1), indicating their importance for vital functions. Indeed, latrophilin-like aGPCRs are also present in invertebrates, e.g., worms and insects (reviewed in [37]) and have a negative impact on animals’ fitness when deleted [40,55,56,57]. Similarly, deletion of members of the ADGRA family, for example GPR124/ADGRA2, cause lethality in mice with various defects of the CNS-specific vascularization and establishment of the blood–brain barrier [58,59]. BAI1/ADGRB1, a member of the ADGRB family, controls macrophage-mediated engulfment of apoptotic cells [60]. Members of the family ADGRC are involved in planar cell polarity and ciliogenesis (reviewed in [61]). Interestingly, severe human diseases have been described for mutations in CELSR1/ADGRC1 and VLGR1/ADGRV1 but not for any member of the families ADGRA, B, D, and L [62]. This may indicate that loss of one of these very conserved aGPCR members has (i) no obvious or disease-relevant phenotype or (ii) is not tolerable with human life. These two hypotheses can be tested by determining the constraint (observed vs. expected mutation rates (o/e ratio)) in a given gene [63]. We have recently shown that the synonymous mutation rate has no significant difference between GPCR genes related to known monogenic diseases and those genes which have not been already associated with monogenic phenotypes [62]. However, loss-of-function (LoF) mutations (stop, frameshift, splice) show a significantly lower ratio between the observed and expected mutation rates in GPCR genes with known disease relevance indicating that, based on the constraint, one can predict candidate GPCR genes leading to severe functional defects upon inactivation [62]. As shown in Appendix A, the constraint of ADGRA, B, C, and L is significant below the average LoF o/e ratio of all aGPCR. This supports the hypothesis that members of these conserved aGPCR families are very relevant for species’ fitness. In contrast, the ADGRE and F families are significantly above the overall average (Appendix A), indicating a reduced constraint compared to other aGPCR families. Indeed, the presence of several members of the E and F families in mammalian and bird genomes is variable and CD97/ADGRE5 and GPR113/ADGRF3 are completely lost in birds (Table 1). This may indicate that the two aGPCRs are not required in this vertebrate class or that the loss of both genes provided an advantage. This is an interesting result because CD97/ADRE5 is one of the best-studied aGPCR with well-established functions in leukocyte homeostasis and apoptosis regulation (reviewed in [64,65]). There is a number of differences in the innate and adaptive immune system between birds and other vertebrates [66] but it remains open, which functions of CD97/ADGRE5 in birds and platypus are compensated by other mechanisms and/or became dispensable. Unfortunately, CD97/ADGRE5-deficient mice have no obvious phenotype [67].

Among the aGPCR families with LoF o/e ratios above average, GPR116/ADGRF5 is an exception showing a low LoF o/e ratio (Appendix A). Consistently, mice deficient for this gene suffer from massive respiratory distress due to profound accumulation of alveolar surfactant phospholipids [68]. The ADGRG family has three human members (GPR64/ADGRG2, GPR114/ADGRG5, GPR126/ADGRG6) with LoF o/e ratios below the average (Appendix A). Inactivating mutations in human and mouse in GPR64/ADGRG2 cause infertility due to congenital bilateral absence of the *Vas deferens* [69,70] and GPR126/ADGRG6 defects lead to lethal arthrogryposis multiplex congenita [71,72].

## 3. Materials and Methods

### 3.1. Retrieval of aGPCR Sequences from Databases

All used cDNA sequences and the corresponding amino acid sequences were obtained from GenBank using NCBIs tblastn [73] with set default parameters and an E-value of 1 × 10^−6^. The amino acid sequences of all known 32 human aGPCR and the amino acid sequence of the mouse EMR4/ADGRE4 served as queries. In the case of partially extracted mRNA sequences from NCBI, the database Ensembl was also searched for the full-length sequence. All sequences retrieved from Ensemble instead of GenBank are marked in Appendix A. To ensure that currently unassigned aGPCRs were retrieved as well, the same process was repeated using the 7TM domain of human secretin receptor–like GPCRs as they are proposed to be descendants of the aGPCR family [11].

In our search, we included a selection of chordate species with an assembled genome (Appendix A). To get a broad representation within the mammalian and avian groups, at least one species from each monophyletic clade [74,75] was included. Within reptiles, one representative member of each order (Testitudines, Crocodylia, Squamata) was selected. From the order of Squamata, two species from different suborders were chosen as this order contains a broad spectrum of species. Within the order of Sphenodontia, no genome of any species fulfilled the requirements to be included.

We applied the same selection process to amphibians and chose at least one species to represent the orders Anura and Caecilia. However, in the order Caudata, there was no species with a fully assembled genome. For fishes, we focused on two species, zebrafish (*Danio rerio*) and pufferfish (*Takifugu rubripes*), which often serve as model organisms since their genomes are continuously curated. An overview of all analyzed species and the corresponding version of their genome annotation can be found in Appendix A.

### 3.2. Alignments and Phylogenetic Analyses

Multiple alignments were generated using the analyzing tool MEGA11 [17,18] performing two different alignment methods. Firstly, we employed the algorithm MUSCLE [76] in default settings and secondly, the algorithm ClustalW with set default parameters [77]. All alignments were reviewed and curated manually.

The evolutionary history of the 7TM domain of aGPCR amino acid sequences was inferred using the Neighbor-Joining (NJ) and the Maximum Likelihood (ML) method based on the Jones–Taylor–Thornton (JTT) matrix-based model implemented in MEGA version 11.0.7 [17,18]. In both cases the Poisson correction method [78] served as the substitution model and the test of phylogeny was performed using the bootstrap method [32] with 1000 replications for the NJ method and 100 replications for the ML method. For the overview phylogenetic trees, rhodopsin orthologs served as an outgroup. All partial phylogenetic trees analyzing only single families of receptors within the aGPCRs used ADGRV1 as an outgroup. To account for input-order bias, similar trees were generated with at least three different randomized alignments. Of note, we found no major differences in the tree structure by changing the input order. Initial trees for the heuristic search were obtained by applying the NJ method to a matrix of pairwise distances estimated using a JTT model.

### 3.3. Selection Analyses

Selection and gene duplication analyses of vertebrate aGPCRs were performed using the webtool Selectome [29]. In brief, nucleotide (codon)-based alignments are used to generate a phylogenetic tree applying Godon [79]. Based on the individual tree, Selectome uses the branch-site model, which estimates different dN/dS values (ω-values) among branches and among sites. Two models are computed: a null model (H0), in which the foreground branch may have different proportions of sites under neutral selection than the background (i.e., relaxed purifying selection); and an alternative model (H1), in which the foreground branch may have a proportion of sites under positive selection [29].

## 4. Conclusions

In this study, we resolved the evolutionary history of aGPCRs by mining public databases, showing that all nine families present in human were already established in the very beginning of vertebrate evolution. This fact enabled us to describe the evolutionary and structural dynamics of individual members of the aGPCR class and to speculate on their physiological relevance in individual species, orders, and classes; as well as their importance in human diseases. Finally, this data has an important impact on revising the nomenclature of the aGPCR class and their hallmark key residues.

## Figures and Tables

**Figure 1 ijms-22-11803-f001:**
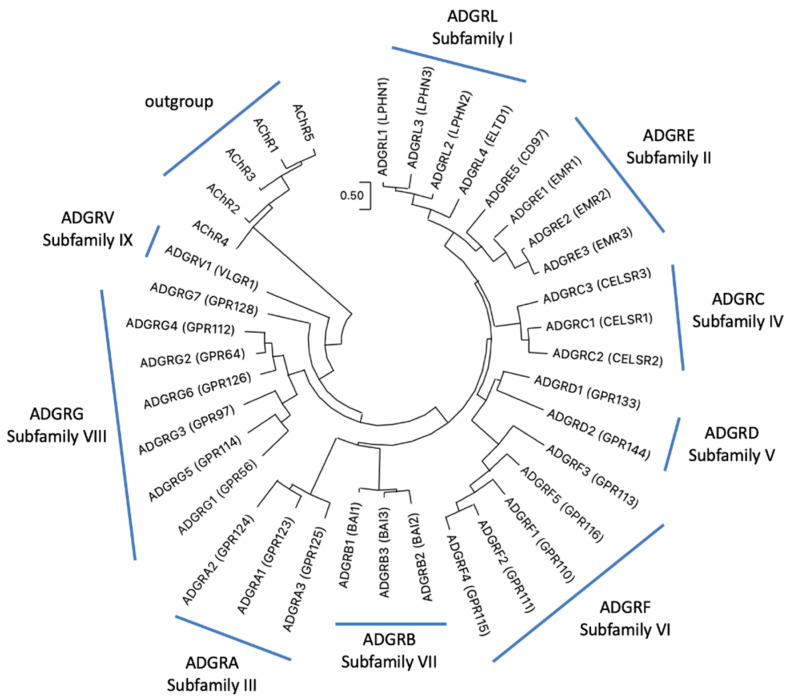
Phylogeny of human aGPCRs showing the current aGPCR subfamilies. The phylogenetic relation of human aGPCRs is shown and nine “subfamilies” were defined [9]. The evolutionary history was inferred by using the Maximum Likelihood (ML) method and JTT matrix-based model [16]. The tree with the highest log likelihood (−16,982.54) is shown. Initial tree(s) for the heuristic search were obtained automatically by applying Neighbor-Join and BioNJ algorithms to a matrix of pairwise distances estimated using the JTT model, and then selecting the topology with superior log likelihood value. The tree is drawn to scale, with branch lengths measured in the number of substitutions per site. This analysis involved human 37 amino acid sequences with the five human muscarinic acetylcholine receptors as outgroup. There were a total of 368 positions in the final data set. Evolutionary analyses were conducted in MEGA11 [17,18]. The accession numbers are given in Appendix A.

**Figure 2 ijms-22-11803-f002:**
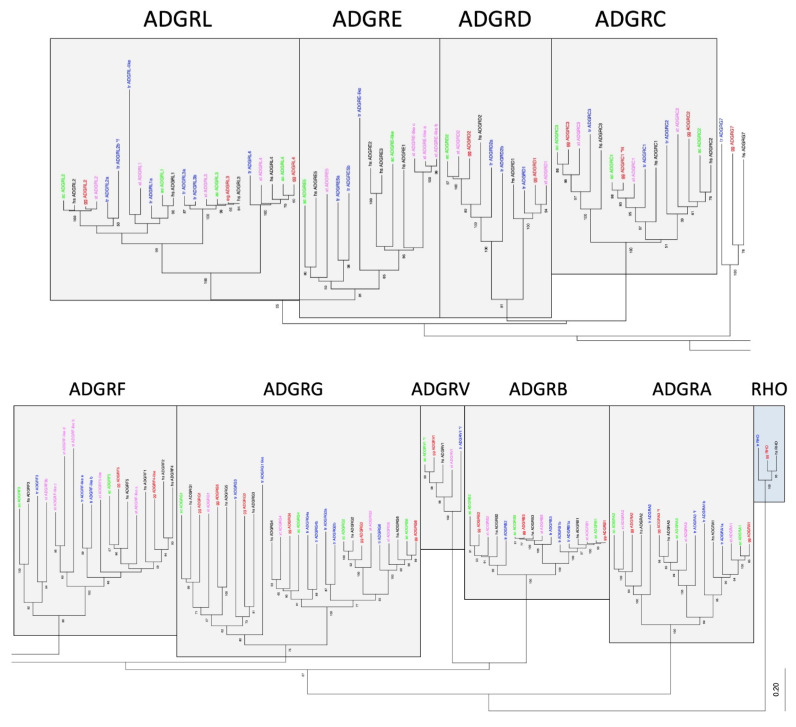
Phylogeny of representative vertebrate aGPCRs. The 7TM amino acid sequence of orthologs of all aGPCR families from human (black, hs: *Homo sapiens*), chicken (red, gg: *Gallus gallus*), lizard (green, ac: *Anolis carolinensis*), frog (magenta, xt: *Xenopus tropicalis*), and bony fish (blue, tr: *Takifugu rubripes*) were aligned using MUSCLE. A Neighbor-Joining tree (NJ) was generated using rhodopsin orthologs from fish, bird, and human as outgroup. The tree was split for better visibility and the currently assigned nine aGPCR families are individually boxed [19]. The percentage of replicate trees in which the associated taxa clustered together in the bootstrap test (100 replicates) are shown next to the branches [32]. The fully resolved NJ and ML trees with bootstrap test (1000 replicates) are given in Appendix A. The accession numbers are given in Appendix A, * f, only sequence fragments were found in the databases.

**Figure 3 ijms-22-11803-f003:**
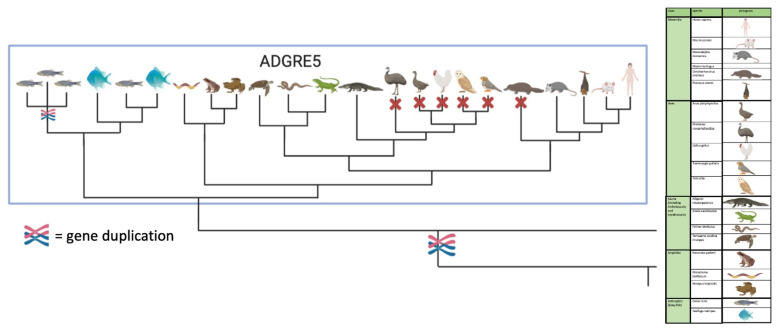
Scheme of the phylogenetic relation of vertebrate CD97/ADGRE5. CD97/ADGRE5 is present in fish, amphibian, reptile, marsupial, and eutherian genomes but not in birds and platypus. The full figure with all members of the ADGRE family is given in Appendix A generated with BioRender.com.

**Figure 4 ijms-22-11803-f004:**
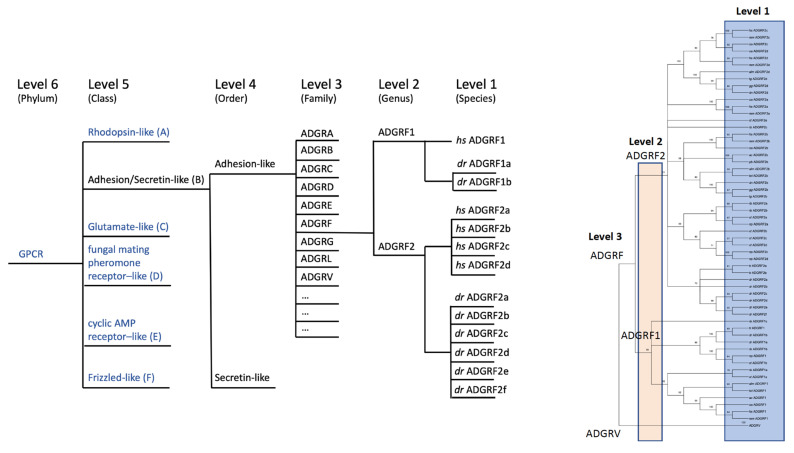
Hierarchy-based nomenclature of aGPCRs. Based on clustering analyses of the 7TM domain the previously suggested nomenclature of aGPCRs [19] was revised. We applied our recently introduced hierarchic level systematics. Level 4 (order) divides secretin-like receptors and aGPCR (ADGR). We extended level 3 (families) to an open nomenclature (ADGRA, B, C, …) where even new families (e.g., ADGRN in other chordates shown in Figure 5) can be newly defined, and assigned level 2 (e.g., ADGRF1, ADGRF2) only to those aGPCRs where a clustering-supported orthology was found between fish and mammals. Individual receptors that descend from level 2 by radiation or duplication are marked by additional lower characters (e.g., *dr* ADGRF1a, *dr* ADGRF1b). *dr*: *Danio rerio*, *hs*: *Homo sapiens*.

**Figure 5 ijms-22-11803-f005:**
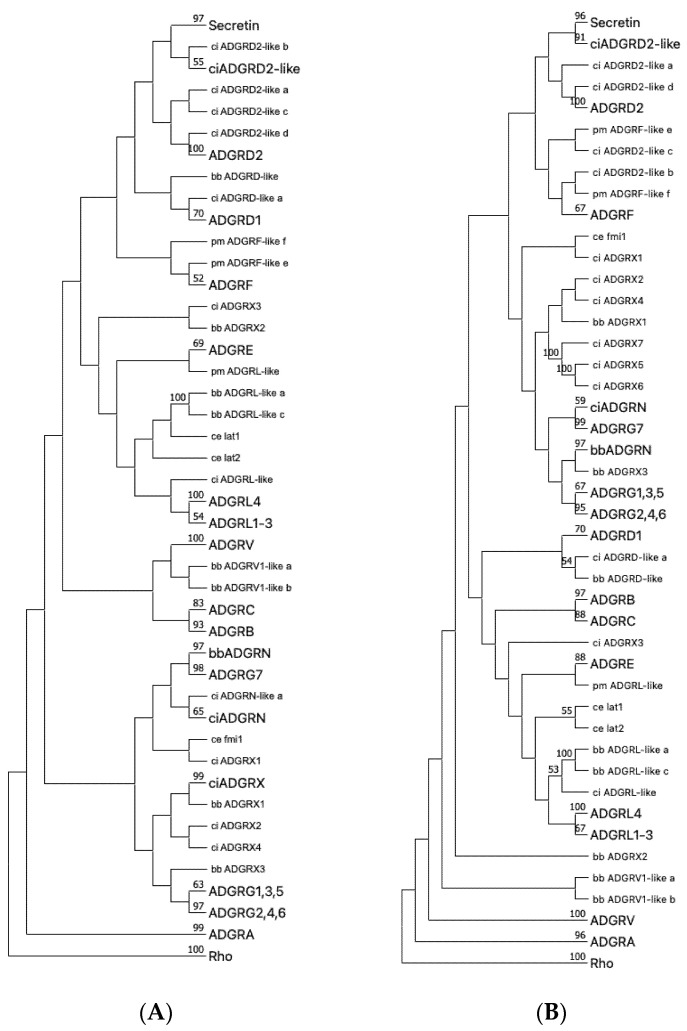
Phylogenetic tree of representative Chordata and *C. elegans* aGPCRs and secretin-like GPCRs. The 7TM domain amino acid sequence of representative *Chordata* and *C. elegans* aGPCRs and secretin-like GPCRs were aligned with MUSCLE (**A**) and ClustalW (**B**). The evolutionary history was inferred using the Neighbor-Joining method [38] based on a sequence alignment of the 7TM part of chordate and *C. elegans* aGPCR orthologs. Rhodopsin orthologs (Rho) served as outgroup. The optimal tree is shown. The percentage of replicate trees, in which the associated taxa clustered together in the bootstrap test (1000 replicates) are shown next to the branches [32]. The evolutionary distances were computed using the Poisson correction method [39] and are in the units of the number of amino acid substitutions per site. This analysis involved 338 amino acid sequences. All ambiguous positions were removed for each sequence pair (pairwise deletion option). Evolutionary analyses were conducted in MEGA [17,18]. The subtrees of currently and newly assigned aGPCR families and the secretin-like receptors were condensed and labeled with a larger font size. *C. elegans*, as a distantly related invertebrate with the well-studied aGPCR members latrophilin 1 and 2 (lat-1, lat-2) and flamingo (fmi) [40], was included to internally evaluated the rooting of the trees. Thus, latrophilins were expected to cluster with vertebrate LPHN/AGDRL (see Figure 1) and flamingo was currently not well-assigned to a vertebrate aGPCR family. Secretin-like receptors are descendants of ADGRD2 as supported by both trees. Uncondensed trees are given in Appendix A. lamprey (pm, *Petromyzon marinus*), lancelet (bb *Branchiostoma belcheri*), vase tunicate (ci, *Ciona intestinalis*), nematode (ce, *Caenorhabditis elegans*).

**Figure 6 ijms-22-11803-f006:**
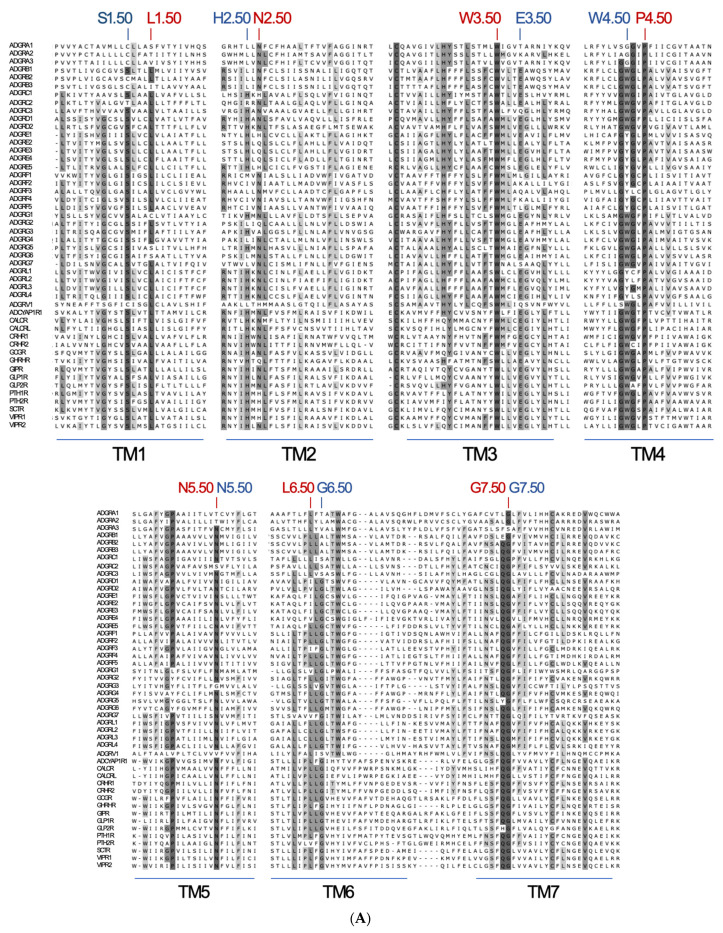
Projection of reference positions and conserved positions in aGPCRs on the cryo-EM structure of ADGRG3/GPR97. (**A**) The amino acid sequence alignment of the seven transmembrane helixes (TM1-7) of human aGPCRs and secretin-like receptors is shown. The previous reference positions (blue) [48] and the newly assigned reference positions (red) are given. (**B**) The newly assigned reference positions (red) and positions which are almost >90% conserved among vertebrate aGPCRs (magenta) are projected into the 7TM domain cryo-EM structure of ADGRG3/GPR97 (PDB: 7D76, [42]). W3.50 (red), P.4.50 (red) and G7.50 (red) are >99% conserved reference positions in vertebrate aGPCRs. The 7TM domain structure is given in two views rotated by 90°.

**Table 1 ijms-22-11803-t001:** Presence of aGPCRs in Chordata and signatures of positive selection in branches. The presence of individual aGPCR members are depicted in dark grey. In case some animal orders lack the individual member, boxes are in light grey. Furthermore, some species lack a clear one-to-one orthology to a human aGPCR member. In such a case, the box spans over more than one human aGPCR. For example, in lamprey there are two members of the family A, but one member has similar identities to the human ADGRA1 and ADGRA3. The complete absence of a member is shown in white boxes. For selection analysis the webtool Selectome (https://selectome.org/; accessed on 15 August 2021) with the default aGPCR ortholog alignments and phylogenetic trees was used [29]. This analysis uses the branch-site model to determine ω-values among branches. The assigned significance of the detection of positive selection on the selected branch can be extracted with the webtool by entering the respective aGPCR gene name. Branches with significant signatures of positive selection are given in black for major branches, in red for fishes, in green for birds and reptiles, and in blue for mammals. Branches marked with * underwent duplication and positive selection.

aGPCR	Old Symbol	Mammals	Birds	Reptiles	Amphibians	Fish	Lamprey	Lancelet	Ciona	Selection in Branch
ADGRA1	Gpr123									Protacanthopterygii
ADGRA3	Gpr125									-
ADGRA2	Gpr124									-
ADGRB1	Bai1									Clupeocephala *, Protacanthopterygii
ADGRB2	Bai2									-
ADGRB3	Bai3									-
ADGRC1	Celsr1									Sarcopterygii, Actinopterygii, Euteleosteomorpha,Percomorpha, Poecilia, Tetrapoda, Boreoeutheria
ADGRC2	Celsr2									Actinopterygii *, Neopterygii, Clupeocephala, Euteleostomi *, Amniota, Passeriformes
ADGRC3	Celsr3									-
ADGRD1	Gpr133			e						Otomorpha
ADGRD2	Gpr144	a								Sarcopterygii, Neopterygii, Osteoglossocephalai *, Euteleostomi *
ADGRE1	Emr1									Cercopithecidae, Ursus
ADGRE2	Emr2									Panthera, Lemuriformes, Marmotini*
ADGRE3	Emr3									-
ADGRE4	Emr4	b								Eutheria *
ADGRE5	Cd97									Otomorpha, Percomorpha *
ADGRF1	Gpr110									-
ADGRF2	Gpr111	c								Hystricomorpha
ADGRF4	Gpr115	d								-
ADGRF5	Gpr116									Archeosauria (Neognathae, Galloanserae, Iguania), Mammalia, Clupeocephala *, Ovalentaria, Euteleosteomorpha, Eupercaria
ADGRF3	Gpr113									Ovalentaria *, Percomorpha
ADGRG1	Gpr56									Archeosauria, Testudinidae, Neopterygii, Clupeocephala, Oryzia
ADGRG3	Gpr97									Neopterygii, Clupeocephala, Oryzia *
ADGRG5	Gpr114									Theria, Archosauria *
ADGRG2	Gpr64									Euteleosteomorpha, Cyprinodontidae, Archeosauria
ADGRG4	Gpr112									Osteoglossocephalai *, Percomorpha
ADGRG6	Gpr126									Laurasiatheria, Osteoglossocephalai, Otomorpha, Protacanthopterygii, Percomorpha, Atherinomorpha
ADGRG7	Gpr128									Oryzias
ADGRL1	Lphn1									-
ADGRL2	Lphn2									-
ADGRL3	Lphn3									-
ADGRL4	Eltd1									Clupeiformes
ADGRV1	VLGR1									-

^a^ Pseudogenization in *Chrysochloris asiatica*, *Loxodonta africana*, *Mus musculus*, *Rattus norvegicus*, *Dasypus novemcinctus*, *Orcinus orca*, *Rhinolophus ferrumequinum*; ^b^ Pseudogenization in Hominidae; ^c^ Gene loss in recent Cetaceans; ^d^ Gene loss in some dolphins; ^e^ Gene loss in *Anolis carolinensis*.

**Table 2 ijms-22-11803-t002:** Proposed nomenclature of human aGPCRs based on phylogenetic clustering in vertebrates. The old and new nomenclatures of human aGPCRs are given. Revised names which are different to the previous annotation [9] are marked in grey. * pseudogene in human.

Human aGPCR Nomenclature V3.0	New aGPCR Family Nomenclature (V3.0)	aGPCR Nomenclature V2.0	Old Symbol (V1.0)
ADGRA1	ADGRA	ADGRA1	Gpr123
ADGRA2		ADGRA3	Gpr125
ADGRA3		ADGRA2	Gpr124
ADGRB1	ADGRB	ADGRB1	Bai1
ADGRB2		ADGRB3	Bai3
ADGRB3		ADGRB2	Bai2
ADGRC1	ADGRC	ADGRC1	Celsr1
ADGRC2		ADGRC2	Celsr2
ADGRC3		ADGRC3	Celsr3
ADGRD1	ADGRD	ADGRD1	Gpr133
ADGRD2		ADGRD2	Gpr144
ADGRE1	ADGRE	ADGRG7	GPR128
ADGRF1	ADGRF	ADGRF3	Gpr113
ADGRF2a		ADGRF1	Gpr110
ADGRF2b		ADGRF5	Gpr116
ADGRF2c		ADGRF2	Gpr111
ADGRF2d		ADGRF4	Gpr115
ADGRG1	ADGRG	ADGRG2	Gpr64
ADGRG2		ADGRG6	Gpr126
ADGRG3		ADGRG4	Gpr112
ADGRG4a		ADGRG3	Gpr97
ADGRG4b		ADGRG1	Gpr56
ADGRG4c		ADGRG5	Gpr114
ADGRL1	ADGRL	ADGRL1	Lphn1
ADGRL2		ADGRL2	Lphn2
ADGRL3		ADGRL3	Lphn3
ADGRL4		ADGRL4	Eltd1
ADGRL5		ADGRE5	Cd97
ADGRL6a		ADGRE1	Emr1
ADGRL6b		ADGRE2	Emr2
ADGRL6c		ADGRE3	Emr3
(ADGRL6d)		(ADGRE4)	(Emr4) *
ADGRV1	ADGRV	ADGRV1	VLGR1

## Data Availability

Not applicable.

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
