# Peer review of "The Evolutionary History of Vertebrate Adhesion GPCRs and Its Implication on Their Classification"

_ijms, 2021, doi:10.3390/ijms222111803_

Round 1

Reviewer 1 Report

The manuscript “The evolutionary history of vertebrate adhesion GPCRs and its implication on their classification# submitted by Wittlake and co-workers describes a study of adhesion GPCRs in various species. The need for classification of this complex group of molecules is obvious and commendable.  Adding data on lower vertebrates and invertebrates provides a more thorough understanding of these molecules demonstrating individual gain or loss of various families in various species.

Major issues:

As it stands the manuscript needs merging of results and discussion to allow a proper evaluation.

Detailed issues:

Line 45-46: “Based on the phylogeny of the human aGPCR genes, nine families (ADGRA, B, C, D, E, F, G, L, V) were defined (Figure 1).» Were these nine families defined based on 33 human sequences alone or aligned to sequences from other species? In Fig.1 a variety of aGPCR sequences are include also supporting this 9 family definition so it becomes confusing.

77-78: The current nomenclature also does not consider the multitude of transcript variants aGPCR genes can encode. This is of high relevance because the derived proteins can substantially differ in structure and function [13].

Comment: Note that genome isotypes are predictions- not supported by true transcripts. True

transcript reports are rare for non-model organisms. And please explain what kind of alternative transcripts are shown in vivo for these receptors exemplifying why this is important? Alternative 5-7 TM domains? Alternative signal/ 5’ or 3’ sequences?

Line 79-81: Furthermore, the assembled vertebrate repertoire of aGPCRs allowed for structural comparison and reveal structural determinants relevant for maintaining the specific functions of aGPCRs.

Comment: Only vertebrates?

Line 81-83: Based on the identification of signatures of positive selection and frequency analysis of loss-of function variants in humans, we evaluated the possible functions of aGPCRs in physiological and adaptive processes and their role in human diseases.

Comment: Can one determine function based on sequence? Do the authors know that a clustering aGPCR in humans and e.g. a teleost have the same functional ligands when they belong to a phylogenetic cluster?

Figure 1. Where are the accession numbers for these sequences? Please refer to this in figure legend. And also highlight the rhodopsin outgroup making it easily visible. Another thing- this figure is now part of the introduction where mammals are key focus but it should be part of results. Change location. And it is very difficult to see what is human, chicken etc sequences. Can the various species names get a unique colour too e.g. gg_ADGRB, gg_ADGRD1 etc all colored green? And the circle format of this figure makes it difficult to compare to the one presented in Scholz et al.(2019). Can it be rectangular instead?

Results

Introduction contains all vertebrates, but then results (2.1 Repertoire of aGPCRs in mammalian classes) only focus on mammalian aGPCRs although table 1 lists the presence of aGPCR members in all vertebrates. Either only list mammalian data in 2.1, make a sharper distinction between the two chapters, or merge 2.1 and 2.2.

Table 1 lists presence or absence of the various aGPCR variants. It is unclear why lamprey has only one box covering both ADGRA1+ADGRA3 etc. Is there only one lamprey sequence with identity to both ADGRA1+ADGRA3? Please include an explanation in figure legend.

Also branches with significant signatures of positive selection  are shown in the right hand column. It is difficult to assess these data without knowing how many species within each branch were included in the study. Are such data shown in supplementary? See also comment below.

Line 133-136: Interestingly, two members of the aGPCR class show a high frequency of pseudogenes across the mammalian linages: EMR4/ADGRE4 and GPR144/ADGRD2 (Table 1), indicating specific functions in some mammalian species but not a vital requirement of these two receptors in mammals.

Comment: When a clade/ sequence is deleted in a species, would other aGPCRs take over that role? Also remark that although a gene does not seem present in a genome, it may have been ignored in the genome assembly.

Line 161-165: Thus ADRGE5 and the group ADGRE1-4 have diverged mainly after tetrapod split from fishes in Devonian [19].

Comment: ADGRE5 has orthologs in mammals, fishes, amphibians and reptiles so did not diverge after tetrapods split from fishes but was lost in birds and platypus.

Figure 2. Guess the duplication is a gene duplication? The left part of this figure shows the species kingdom with species latin names and species abbreviations, but the abbreviations are not used in the figure only the pictograms. Should this figure be presented earlier also explaining figure 1 species abreviations?

Figure 3 legend: Species abbreviations are as follows: lamprey (pm, Petromyzon marinus), lancelet (bb Branchiostoma belcheri), vase tunicate (ci, Ciona intestinalis), nematode (ce,Caenorhabditis elegans)

Line 225-226: It should be noted that we did not find any new aGPCRs in vertebrates not related to the already known aGPCR families in mammalian genomes.

Comment: would the current analysis pipeline identify aGPCR sequences that diverge significantly from known mammalian sequences? And in Line 278-280 the authors state that “Interestingly, we found two currently not assigned aGPCR families in C. intestinalis and lancelet (referred to as ADGRN and ADGRX, Figure 4), which are not present in vertebrates.” So present these invertebrate sequences here, but refer to 2.5 for additional data.

Line 241-244: As expected, most significant w-values were found in fish aGPCR orthologs (Table 1), where after duplication one branch evolved under positive selection, whereas the other homolog remained under purifying (negative) selection. There were also a few cases in tetrapods, in which gene duplication was followed by positive selection of one copy.

Comment: these values are presented in Table 1, but very hard to see. They should be include in a separate table under section 2.4 specifying each species where selection was observed.

Line 258-259: They may indicate that members of type i) mainly participate in maintaining conserved physiological function and of type ii) contribute to adaptive functions.

Comment: Do the receptors adapt to sequence variation in ligands?

Figure 4. Phylogenetic tree of representative Chordata and C. elegans aGPCRs.

Comments: Are the entire data (uncondensed) presented in supplementary? The Figure title states Chordata and C.elegans aGPCRs, but also includes secretin which is or is not a member? Andf where does that secretin originate from? Also the text talks about multiple secretin sequences but figure only shows one? Label secretin sequence(s) with different color or is there only one? Latrophilins are presented in introduction with alternative names but here as lathrophilins making it confusing. Flamingo has not been introduced at all. Change name in Fig.4, introduce in introduction, figure legend, delete or explain relevance more clearly?

Line 336-337: This may indicate that secretin-like GPCRs have evolved from parts of the 7TM domain of different aGPCRs.

Comment: How- through recombination perhaps?

Figure 5A: It is impossible to relate to this figure due to small font. Increase the font by dividing the figure into two paragraphs.

Figure 5B. What are these two figures? Not mentioned in legend.

Discussion:

This section is divide into paragraphs with repetitions of data presented in results and also introducing new aspects not being discussed previously (Stachels?). Please merge Results and Discussion, delete duplicates between results and discussion and shorten the discussion section. The discussion also contains results (Figure 6 and Table 2).

Commenting on the discussion part needs to await merging with results.

Author Response

Please check it in the attachment, thank you

Reviewer 2 Report

The manuscript describes recent findings in the aGPCRs classification and appearance in various species. The classification has been updated concerning the previous one based on human aGPCRs (Ref. 8 and 12) based on data from newly described genomes. I don’t have any major comments except a few minor mentioned below:

  • lines 64-70 – I would rather use the SCOP structural classification (classes/folds/superfamilies/families). Mixing various terms makes GPCR publications difficult to read by non-experts in the field.
  • I would add more on what was stated in lines 81-84
  • it would be good to emphasize findings on aGPCRs as ancestors of secretin GPCRs to improve novelty regarding ref. 12
  • Line 104 – ‘data sets are available upon request’, it should be included in supplementary files as the basis of this study.
  • Figure 2 – right panel not visible. Figure could be extended to the full page instead of putting it to the supplementary.
  • Figure 5 – alignment not visible, should be moved to supplementary or reformatted.
  • it would be good to add a description of functional implications of aGPCRs genes loss in some species at least in a few cases.
  • why Authors did not use ClustalOmega instead of ClustalW, which is a bit old program? But in general, no further comments on methodology.

Author Response

(The authors gave the same response as above.)

Round 2

Reviewer 1 Report

The revised version of this manuscript has greatly improved and it is now ready for publication.